



# Observations of sunlit $N_2^+$ aurora at high altitudes during the RENU2 flight

Pål Gunnar Ellingsen[1], Dag Lorentzen[2,3], David Kenward[4], James H Hecht[5], J. Scott Evans[6], Fred Sigernes[2,3], and Marc Lessard[4]

[1] Department of Electrical Engineering, UiT The Arctic University of Norway, 8505 Narvik, Norway
[2] Department of Arctic Geophysics, the University Centre in Svalbard, P.O. Box 156, 9171 Longyearbyen, Norway
[3] Birkeland Centre for Space Science, Department of Physics and Technology, University of Bergen, 5020 Bergen, Norway
[4] The University of New Hampshire, Durham, NH 03824, USA
[5] Space Science Applications Laboratory, The Aerospace Corporation, El Segundo CA, USA
[6] Computational Physics Inc., Springfield, VA, USA

**Correspondence:** Pål Gunnar Ellingsen (pal.g.ellingsen@uit.no)

**Abstract.** We present measurements of sunlit aurora during the launch of the Rocket Experiment for Neutral Upwelling 2 (RENU2) on the 13th of December 2015 at 07:34 UT. The *in situ* auroral conditions coincide with those of sunlit aurora, and were characterised by the 391.4 nm and 427.8 nm $N_2^+$ emissions. A correlation between several auroral wavelengths, as measured by a meridian scanning photometer was used to detect sunlit aurora and indirectly neutral upwelling. These results, based on ground data, agree well with the RENU2 measurements recorded during its pass through the sunlit polar cusp. Using data from RENU2 and the solar photon flux, it was found that sunlit aurora was a major part ($\approx 40\%$) of the observed 427.8 nm emission.

## 1 Introduction

Upwelling of neutrals and ions is known to occur in the cusp region of the auroral oval (Lühr et al., 2004; Lorentzen et al., 2007; Carlson et al., 2012; Sadler et al., 2012). The upwelling is known to significantly increase the density of the upper ionosphere, and can be observed by proxy as an increase in satellite drag over the cusp (Lühr et al., 2004). The underlying mechanisms behind the increased density above 400 km are not fully understood, and several explanations have been proposed. Carlson et al. (2012) used a first-principle physics model to show an increase in cusp and polar cap densities due to plasma flow shear through soft electron precipitation. According to a theoretical study by Deng et al. (2013), the precipitation of soft electrons together with a Poynting flux contributed to an increase in neutral density at 400 km. The electrons contributed to an increase in the Pedersen conductivity in the F-region. Electrons with around 100 eV of energy dominated this process. Cohen et al. (2015) used a model to study the relation between electron temperature and density, and ion upwelling. They found that auroral precipitation, electron temperature and density increases all result in an increased upwelling. Due to the localised nature of the upwelling and lack of good techniques for remote measurement of neutrals in the ionosphere, sounding rockets have been used to gather *in situ* observations.





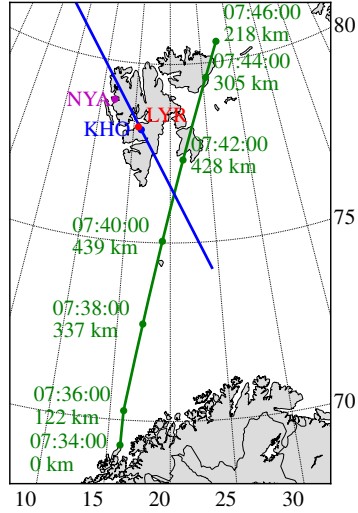

**Figure 1.** Figure showing the ground track for the RENU2 rocket, green, and the Meridian Scanning Photometer (MSP) scan line, blue, when scanning above 20° elevation angle. The location of Longyearbyen (LYR), Ny-Ålesund (NYA) and the Kjell Henriksen Observatory (KHO) are shown.

The Rocket Experiment for Neutral Upwelling 2 (RENU2) was launched from Andøya Space Centre (ASC) on the 13th of December 2015 at 07:34 UT (≈10:34 MLT). It flew through a sun illuminated cusp over Eastern Svalbard. The launch track is shown Figure 1. The payload carried a large variety of instruments for measuring the *in situ* magnetic and electric fields, ion and electron density and energy, optical emissions and more. Details about the sounding rocket and its instrumentation can be found in the overview article for RENU2 (Lessard et al., 2019).

During the RENU2 campaign ground-based support was available from Svalbard, both from Longyearbyen (LYR) and Ny-Ålesund (NYA), see figure 1. Support from the SuperDARN network of HF radars was also given (Greenwald et al., 1985). In addition to collecting data for post-launch analysis, data was used as input for the launch decision. In this paper we will concentrate on optical ground-based data from LYR site, the Kjell Henriksen Observatory (KHO, Geographical coordinates: N 78.148°, E 16.043°, magnetic coordinates (2015): N 75.55°, E 109.66°) in combination with rocket data, and focus will be studying the link between sunlit aurora and neutral upwelling, and how different processes contribute to the $N_2^+$ emissions.

## 2 Theory

### 2.1 Nitrogen ionisation and emission

Cusp aurora is optically characterised by an irradiance ratio I(630.0nm/557.7nm) > 1, and a typical electron precipitation energy of ≈ 0.5 keV. The Kjell Henriksen Observatory is placed directly beneath the statistical dayside auroral oval, with a negative solar zenith angle for more than two months of the year. However, parts of the ionosphere will always be sunlit in the





daytime. The cusp auroral red emission line - 630.0 nm [OI] - has an average emission altitude of $\approx 250$ km (Johnsen et al., 2012) due to collisional quenching at lower altitudes. In addition to the atomic oxygen emission, emissions in the first negative band of the $N_2^+$ ions have previously been observed in the sunlit part of the ionosphere at Svalbard (Deehr et al., 1980). During
the RENU2 launch, the onboard photometer observed the 391.4 nm $N_2^+$ 1N (0,0) emission, while the ground based Meridian Scanning Photometer at KHO observed the 427.8 nm $N_2^+$ 1N (0,1) emission.

The ionised nitrogen in the auroral $N_2^+$ 1N band is normally produced by direct ionisation via electron impact

$$N_2 + e \longrightarrow N_2^+ + e + e'$$

where $e'$ is a thermal electron.

However ionised nitrogen can also be produced from molecular nitrogen by solar extreme ultra violet (EUV) radiation via (Hunten, 2003; Jokiaho, 2009)

$$N_2 + h\nu(< 79.6\ nm) \longrightarrow N_2^+ + e,$$

where $h\nu$ is the EUV photon.

Under sunlit conditions in the ionosphere, the $N_2^+$ ion in its ground state (X), can experience resonant absorption of solar
photons, ending up in an excited state ($N_2^+$ A or B) and thus contributing to emissions from the Meinel or 1N band, respectively. The more common way of exciting $N_2^+$ is through precipitating electrons. In sunlit aurora it is expected that both will occur, while in non sunlit condition, only precipitating electrons will excite the $N_2^+$.

## 2.2   EUV illumination of the ionosphere

At the time of the rocket launch, the area North of the launch site (ASC) was experiencing polar night, with the Sun continuously
below the horizon, as seen from the ground. Higher altitudes had some sunlight, though the amount of EUV radiation depends on what part of the atmosphere the solar radiation had propagated through (Cohen et al., 2015). The penetration depth of EUV is wavelength dependent, and the majority of the EUV spectrum is attenuated by at least $1/e$ below 150 km (Brasseur, 1986). Therefore we will define a sunlit ionosphere as an ionosphere where the incident EUV has not propagated lower than 150 km before reaching the point of interest.

In this article the definition of sunlit aurora will be the same as the one by Hunten (Hunten, 2003). In this summary he specifies that sunlit aurora occurs when $N_2^+$ fluoresces in the blue and violet as a result of being excited from the ground state by higher energy sunlight.

## 2.3   Polward moving auroral forms and cusp processes

It is known from literature that Poleward moving auroral forms (PMAFs) are associated with cusp aurora processes such
as neutral and ion upflows (Moen et al., 2004), polar cap patches (Lorentzen et al., 2010) and flow channels (Sandholt and Farrugia, 2007; Herlingshaw et al., 2019). PMAFs are ionospheric signatures of pulsed magnetopause reconnection (Fasel, 1995; Sandholt and Farrugia, 2007). The upflows present under PMAFs can allow for the generation of sunlit aurora due to the



increased availability of nitrogen at higher altitudes. If the ionosphere is illuminated by solar EUV (subsection 2.1), nitrogen
can be ionised, making it available for resonant scattering of sunlight. This gives rise to one of the more commonly studied
sunlit aurora bands, the first negative band of $N_2^+$, and then especially its emission at 427.8 nm (Chakrabarti, 1998; Hunten,
2003). Another sunlit emission is the helium 1.083 $\mu$m line (Harrison and Cairns, 1969; Chakrabarti, 1998), which lately has
not been studied that extensively.

## 3  Observations

The ground based optical observations were done under partly overcast conditions, with some light snow in the air. These
observations were done with several types of optical instruments, but in this work we will concentrate on the meridian scan-
ning photometer (MSP), and a $180°$ fisheye all-sky camera. From the RENU2 payload, measurements from the electron plasma
sensor (EPLAS) and the three photometers will be utilised. Measurements of incident EUV were acquired from the EVE exper-
iment on the Solar Dynamics Observatory satellite. Using the OMNI solar wind data service, it was seen that the interplanetary
magnetic field component, $B_z$, turned southward at about 06:00 UT, and stayed negative until about 13:00 UT, with a weakly
negative value of $-1.5$ nT throughout the time period.

### 3.1  Meridian scanning photometer

The Meridian Scanning Photometer (MSP) scans the magnetic meridian, see figure 1, at three different wavelengths; 630.0 nm,
557.7 nm and 427.8 nm. The two first wavelengths are OI lines, while the last is the previously mentioned $N_2^+$ line. Narrow
interference filters, with a bandpass of $\approx 0.4$ nm, are mounted in individual tilting filter holders. By tilting the filter away from
the measured emission line, the background can be measured and subsequently subtracted. Each revolution of the scanning
mirror takes 4 s. One background subtracted scan consequently takes 8 s. Two of these background subtracted scans are
averaged to give a complete acquisition time of 16 s.

Figure 2 shows 3 hours of MSP data centered around the time of launch (07:34 UT). Prior to the launch a number of
poleward moving auroral forms (PMAFs) are observed. The first ones are seen around 06:30 UT. PMAFs are indicative of
pulsed magnetopause reconnection (i.e flux transfer events) (Sandholt and Farrugia, 2007; Russell and Elphic, 1978). The
payload flew East of KHO, and through the extended PMAF located on the keogram between 07:40 UT (360 s) and 07:45 UT
(540 s) and 120 to 60 degrees scan angle.

### 3.2  All-sky camera

The all-sky camera in use during the rocket campaign was a Sony a7s equipped with a 180 degree fisheye lens (Sigma 8mm
f/3.5 EX DG Circular Fisheye). It is a full format CMOS camera, and was taking images every 30 s with a 5 s exposure. During
the launch there was light snow settling on the dome of the camera (cleaned periodically). The images collected were sufficient
to see that during the launch there was a clear arc of dayside aurora overhead, extending from magnetic east to magnetic west.
A video (Sony_launch.mp4) of these images can be found in the supplemented material.





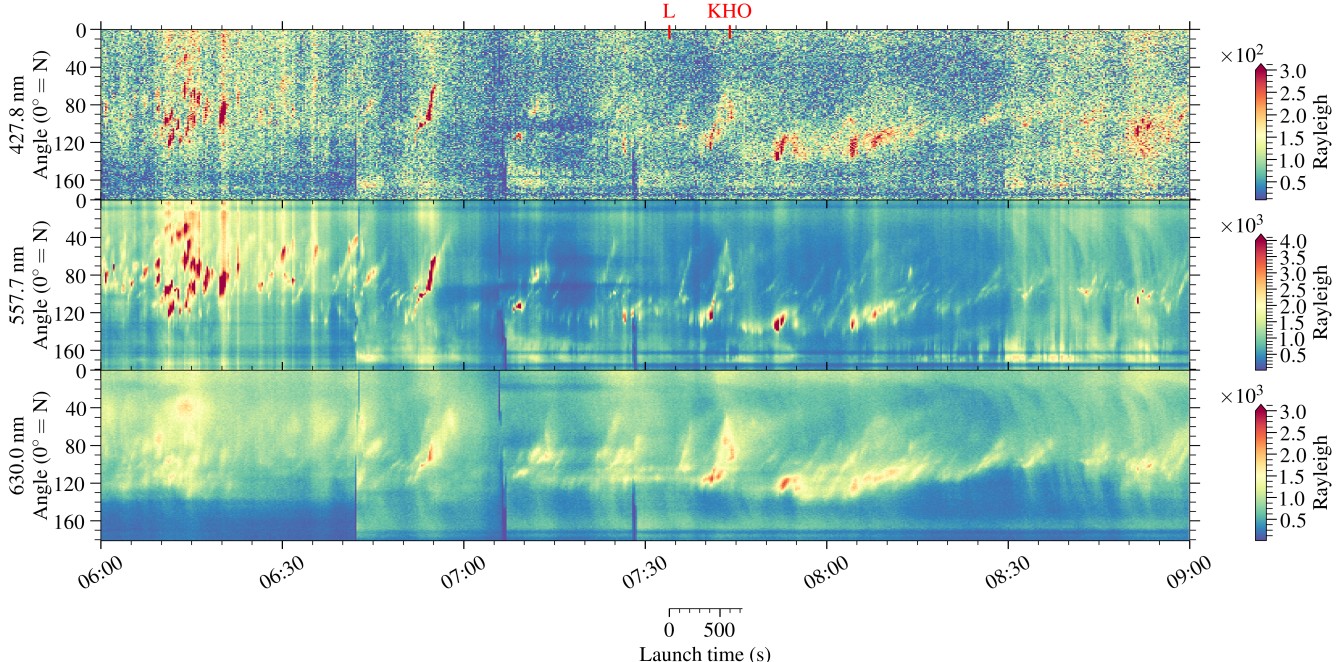

**Figure 2.** The MSP data from 6 UT to 9 UT. It shows the three recorded wavelengths over the North to South scan along the magnetic meridian. The vertical dark blue stripes are from when the slit covering the MSP was cleaned of snow. *L* and *KHO* mark the launch and passing east of KHO.

### 3.3 RENU2 photometers

RENU2 was equipped with a three-channel photometer measuring the red 630.0 nm OI emission, the green 557.7 nm OI emission and the 391.4 nm $N_2^+$ 1N (0,0) emission. Details of the instrumentation can be found in (Lessard et al., 2019). In this work, the focus will be on the nitrogen channel, which is shown in figure 6. Observations and analysis from the other two channels is available in the paper by Hecht et al. (2019).

### 3.4 RENU2 Electron Plasma Sensor and EVE Solar Dynamics Observatory

In this work, we utilize two instruments; the Electron Plasma Sensor (EPLAS) detector onboard the RENU2 rocket and the Extreme Ultraviolet Variability Experiment (EVE) onboard the Solar Dynamics Observatory (SDO) satellite (Pesnell et al., 2012). EVE contains two spectrographs; MEGS-A (6–33 nm) and MEGS-B (33–105 nm). Unfortunately MEGS-A broke on the 2014-05-26, leaving only MEGS-B working. In figure 3 the average irradiance on the day of the rocket (dashed orange), in addition to the closest day when the MEGS-A was running (blue line) can be seen. The closest match was calculated by

correlating the valid parts of the spectra together. In the figure the spectral lines which are part of the solar EUV spectrum can be seen on top of a dimmer continuous spectrum.

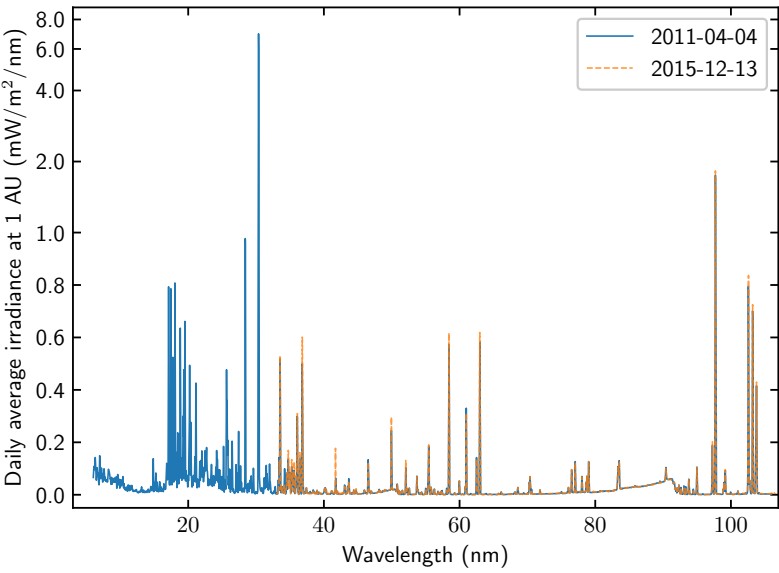

**Figure 3.** Measurements of daily average irradiance at 1 AU from the EVE instrument from the day of the launch 2015-12-13 (orange) and the closest match to the launch day from when the MEGS-A detector worked (blue). Notice that the irradiance scale is linear below 1 mW/m$^2$/nm and logarithmic above.

The Electron Plasma (EPLAS) instrument is a tophat electrostatic analyzer which measures electron energy from 5 eV to 14.6 keV around a 360 degree field of view with 10 degree pitch angle resolution and 42 ms time resolution. The energy steps are logarithmic, and each step has 1 ms dwell time. Further details can be found in Kenward et al. (2019). Figure 4 shows the measurements taken when the rocket passed through the cusp. In the left plot the characteristic energies of the electrons can be seen. The values are above 50 eV for the whole duration, with some short bursts reaching energies above 200 eV. In the right plot the electron flux is shown. It shows a significant increase in flux at 550 s (07:44:10 UT) which subsides again at 630 s (07:45:30 UT). The increase is not constant within this time period, but comes in both shorter and longer bursts.

## 4   Analysis and discussion

### 4.1   Auroral measurements in sunlit aurora

We will now investigate if the ground based 428.7 nm N$_2$$^+$ emissions observed during the sounding rocket launch can be classified as sunlit aurora, and if the ionisation is driven by solar EUV or precipitating electrons. In order to find the contribution of the different processes, we first start by looking at the ionisation rates. Calculating these rates requires knowledge of the amount of EUV flux at a given height and the particle flux of precipitating electrons. RENU2 reached the Svalbard latitudes





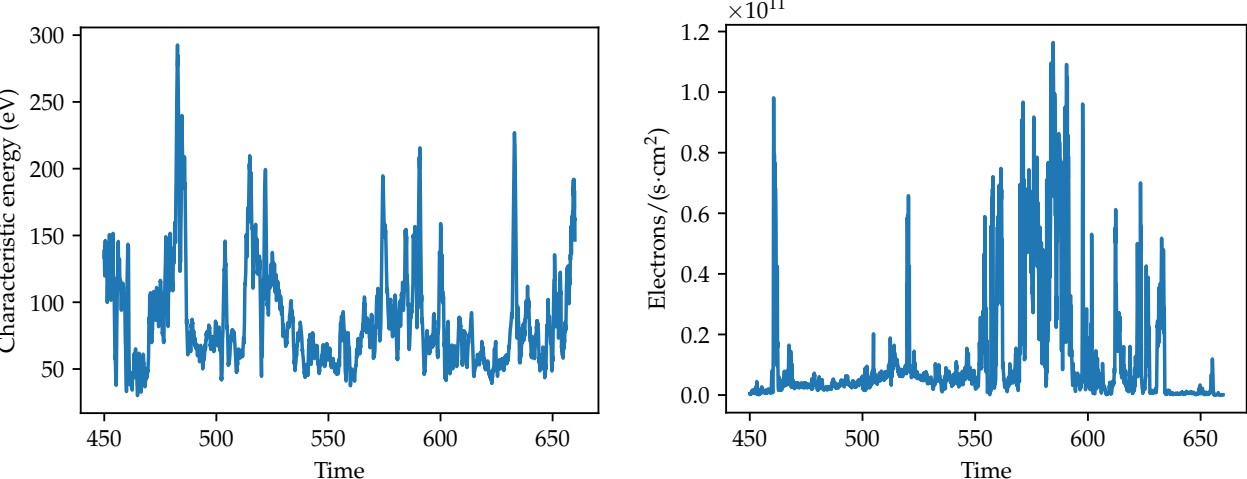

**Figure 4.** EPLAS characteristic energies (left) and electron flux (right) for the part of the flight where the rocket was in the cusp. The x-axis is in flight time with 450 s corresponding to 07:41:30 UT and 650 s to 07:44:50 UT. For a more detailed data set, see Kenward et al. (2019).

at around 07:41 UT (420 s) on the 13th of December 2015 At that time the solar depression angle was 15 degrees, and the ionosphere was sunlit above $\approx 223$ km altitude (per definition in subsection 2.2). The sounding rocket was, at that time, close to its apogee altitude of $\approx 450$ km, and thus well within the sunlit part of the ionosphere.

### 4.1.1   EUV photon flux

The EUV flux incident on atmosphere can be obtained from the Extreme Ultraviolet Variability Experiment (EVE) (Woods
et al., 2012; Hock et al., 2012; Didkovsky et al., 2012) on the Solar Dynamic Observatory (SDO) satellite (Pesnell et al., 2012). Using the Level 3 daily average irradiance data (downloaded in 2017), it is possible to calculate the total irradiance in the wavelength range 33–79.6 nm at 1 AU. On the 13th of December 2015, (see figure 3) it is found to be 0.83 mW/m$^2$. After 2014, only the MEGS-B instrument on the SDO EVE platform - which detects wavelengths above 33 nm - was working. Prior to its failure in 2014, the MEGS-A measured between 6–33 nm. As there are some strong spectral lines below 33 nm, a
calculation of the irradiance needs to include the region down to 6.0 nm. In order to extrapolate the incoming EUV irradiance to the original measurement region, the complete merged Level 3 data from the EVE instrument over the total spectrum from 6.0 nm to 79.8 nm was averaged over the complete time series for the instrument (2010–2017). Thus, by comparing the regions below and above 33 nm with the total average, it was found that the total irradiance for the full wavelength range of 6.0–79.8 nm is $\approx 4\times$ the irradiance above 33 nm, with a standard deviation of 1. To check this factor the spectrum closest to the one on the
13th was found by correlating the valid parts (33–79.6 nm) of the spectrum with the daily averages from before the MEGS-A broke. The highest correlation was found with 4th of April 2011, shown in figure 3, with a correlation coefficient of 0.9937.





This spectrum had a factor of 4.03. Taking both approaches into account, a factor of 4 for the time of the RENU2 launch will be used, yielding a total EUV irradiance of 3.3 mW/m$^2$ at a distance of 1 AU.

By using the calculated EUV irradiance, the photon flux can be found using:

$$j_\gamma = \frac{I}{E_\gamma} = \frac{I\lambda}{hc} \tag{1}$$

where $I$ is the irradiance, $E_\gamma$ the photon energy, $\lambda$ the wavelength, $h$ the Planck constant and $c$ the speed of light. To get an estimate, we can set $\lambda = 50$ nm based on the ionisation rates (Itikawa et al., 1986) and use the calculated irradiance $I = 3.3$ mW/m$^2$ to get $j_\gamma \approx 8.3 \cdot 10^{10}$ cm$^{-2}$s$^{-1}$.

### 4.1.2 Electron flux

The particle flux of precipitating electrons can be obtained using the Electron Plasma Sensor (EPLAS) on the RENU2 payload, for more details see (Lessard et al., 2019; Kenward et al., 2019). EPLAS measured electron energy, flux and pitch angles for electrons with energy between 10 eV and 15 keV. From the measurements shown in figure 4 the average electron flux was found to be $j_e \approx 1.2 \cdot 10^{10}$ cm$^{-2}$s$^{-1}$ in the time period between 07:41:30 UT (450 s) and 07:44:50 UT (650 s) when the rocket was in the cusp. The characteristic energy for these electrons were between from 50 eV to over 100 eV (with peaks of up to 155 300 eV).

### 4.1.3 Ionisation rates

The last parameters needed to calculate the ionisation rates, are the cross sections for ionisation of N$_2$. From literature it is found that the photoionisation cross section for N$_2$ is $\sigma_\gamma \approx 2 \cdot 10^{-17}$ cm$^2$ (Itikawa et al., 1986). The electron impact ionisation cross sections on N$_2$ in the range around 100 eV is $\sigma_e \approx 2 \cdot 10^{-16}$ cm$^2$ (Bug et al., 2013; Tabata et al., 2006). Using these 160 together with the photon and electron fluxes found in the previous subsections, the photoionisation rate is found to be

$$q_\gamma = j_\gamma \cdot \sigma_\gamma = 1.7 \cdot 10^{-6} \text{ s}^{-1}, \tag{2}$$

while the electron ionisation rate is found to be

$$q_e = j_e \cdot \sigma_e = 2.4 \cdot 10^{-6} \text{ s}^{-1}. \tag{3}$$

These numbers indicate that the EUV can contribute up to 40% of the total production of N$_2$$^+$ at the time of measurement, 165 assuming steady state conditions where there is enough N$_2$ available for ionisation. The two rates being comparable (within the limits of the estimate given above), means that in solar EUV conditions one should be careful to classify all the N$_2$$^+$ as having been produced by the EUV, as an increase in the precipitating electrons can drive the production of N$_2$$^+$ as well. Some of the produced N$_2$$^+$ is already in an excited state, as the cross section for photoionisation into the B$^2\Sigma_u$$^+$ state, is $\approx 2 \cdot 10^{-17}$ cm$^2$ for photons with wavelengths in the 50 nm region (Itikawa et al., 1986). The corresponding cross section for electrons at 100 eV 170 is $\approx 5 \cdot 10^{-18}$ cm$^2$ (Itikawa et al., 1986). These estimated rates are only a small part of the complicated photochemistry of N$_2$$^+$ (Abdou et al., 1984).



A process not included, which is important for the 427.8 nm emission, is the resonant scattering of the sunlight. Remick et al. (2001) estimated the enhancement from resonant scattering at $\approx 50\%$ compared to non sunlit conditions. The exact amount is hard to determine as the Remick et al. estimates could include water absorption, Rayleigh scattering and other uncertainties (Remick et al., 2001).

### 4.1.4 Availability of $N_2$

In the previous section we estimated the ionisation rates of $N_2$ through two different processes. These rates require there to be enough $N_2$ available at the point of ionisation. Determining the amount of available $N_2$ at a given altitude is challenging, especially in the cusp. A baseline can be found using the NRLMSISE-00 atmospheric model for the location and time of interest. If we input the location above KHO and time of the launch, the model returns the profiles shown in figure 5. These model results show that the concentration of molecular nitrogen above 400 km is at least two orders of magnitudes smaller than that of atomic oxygen. As mentioned the model is not correct in the cusp, as there is an increase in the density of neutrals

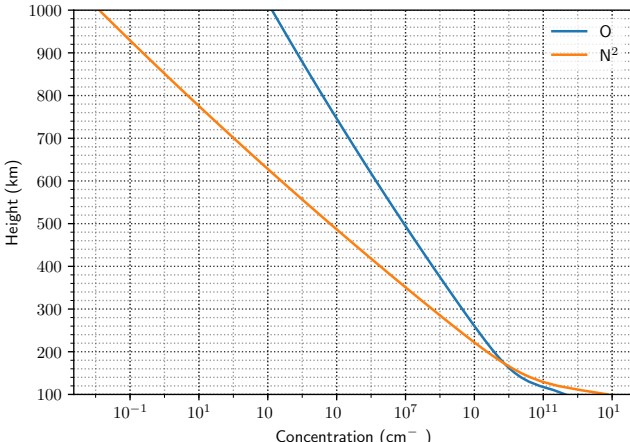

**Figure 5.** An NRLMSISE model of the nitrogen and atomic oxygen content above the KHO observatory in Longyearbyen for the time of the launch (UT 7.5). From the ccmc modelweb, *https://ccmc.gsfc.nasa.gov/modelweb/models/nrlmsise00.php*, M. Picone, A.E. Hedin and D. Drob, Naval Research Laboratory. (Picone et al., 2002).

at these higher altitudes, due to heating and ion upflow (Carlson et al., 2012; Sadler et al., 2012; Lessard et al., 2019). Such an increase provides a significant amount of available neutral nitrogen for ionisation. Under these conditions the model predictions can be viewed as a lower bound for the conditions.

### 4.1.5 Auroral observations from the RENU2 photometers

Hecht et al. (2019) report observations from the redline and greenline photometers onboard the RENU2 rocket. These data are accompanied by a model which includes dayglow and auroral precipitation, the latter based somewhat on the onboard electron





spectrometer data. Also onboard RENU2 was a third photometer that measured emission from the (0,0) band of the $N_2^+$ 1NG

system at 391.4 nm (blue emission) that arises from direct electron precipitation and resonance scattering from sunlit $N_2^+$ ions

that are produced by the precipitation (Romick et al., 1999).

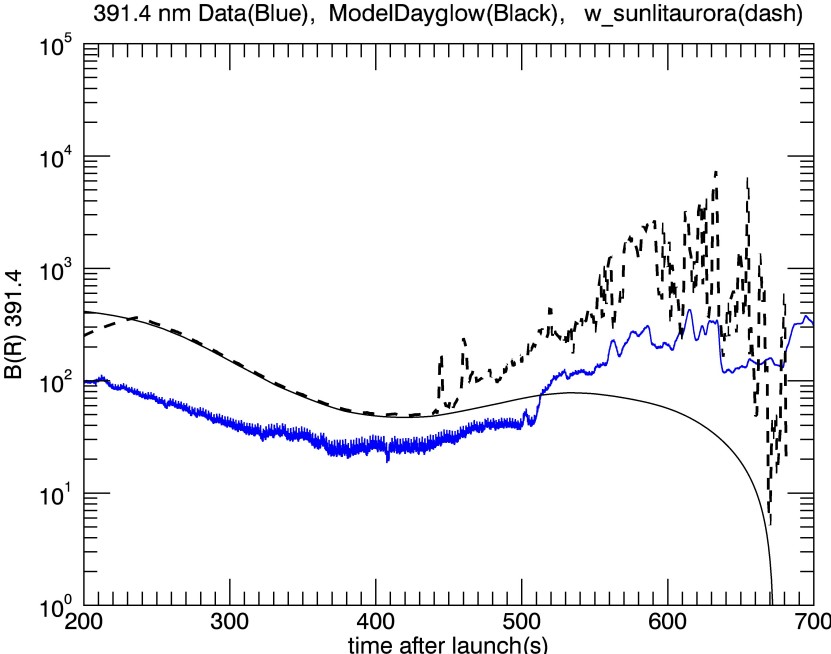

**Figure 6.** A plot of the brightness in Rayleighs of the emission from the (0,0) band of the $N_2^+$ 1NG system at 391.4 nm from the RENU2 blue photometer vs time after launch. Also shown is the model dayglow data (solid line) and the dayglow plus the emission due to the auroral precipitation (see text). 200 s corresponds to 07:37:30 UT and 700 s to 07:45:40 UT.

Figure 6 shows the RENU2 results where the model data is in the same format as in Hecht et al. (2019). Note that the model

is not adjusted for any composition changes although it does account for the portion of the blue emission due to ion production

during precipitation and the subsequent resonant scattering of sunlight. The resonance scattering emission vastly exceeds the

direct auroral emission. The data should therefore reflect auroral precipitation at about 450 seconds (07:41:30 UT) which

continues until the end of the flight. As noted in Hecht et al. (2019) the short-lived nature of some of the intense precipitation

could affect some of the observations of the auroral emissions. Despite this some clear auroral signals are seen: for example,

between 550 and 600 seconds (07:43:10-07:44:00 UT) there are clear increases that correspond to the model predictions of

increases in the blue emission due to resonance scattering of $N_2^+$ ions.

However, the most noticeable aspect of the data is the sudden increase in signal just after 500 seconds (07:42:20 UT) that

continues, as a plateau, until a decrease just around 630 to 640 seconds (07:44:30-40 UT). This is not reflected in the model nor





as shown in Hecht et al. (2019) in the greenline and redline data. Since the model does include sunlit $N_2^+$ this plateau is almost surely due to an excess of these ions above the rocket that begins after about 500 s when the rocket is descending below 400 km. This excess should be an indicator of some ion upflow, above the rocket, associated with the auroral precipitation region, such has been discussed in other studies (Romick et al., 1999; Yau et al., 1993). The decrease at 630-640 seconds is probably due to a boundary associated with the end of the precipitation.

### 4.2 Correlation analysis

In order to investigate if the 427.8 nm emission corresponds to sunlit aurora, it is necessary to look at the emission altitude. For oxygen emissions, the altitudes are related to the energy of the precipitating electrons, resulting in expected altitudes of 100–120 km for 557.7 nm and 220–300 km for 630.0 nm (Omholt, 1971). The 427.8 nm emission is dependent on the availability of ionised nitrogen. From the model results in figure 5 one can infer that the nitrogen emissions would originate at lower altitudes. These would correspond to the altitude of the 557.7 nm emission or lower. This contrasts to sunlit aurora, where the expected emission altitude depends on which part is sunlit above 150 km. This corresponds to the altitude of the 630.0 nm emission. As all of these emissions originate from particle precipitation, one can correlate the emissions from these three lines, in an attempt to determine if the 427.8 nm emission is more correlated with the 557.7 nm or the 630.0 nm emissions.

The correlation will be performed using the MSP data, by calculating the Pearson's correlation coefficient (Edwards and Penney, 1982) between the different wavelengths recorded in one scan. The coefficient is given by

$$\rho_{\lambda_1,\lambda_2}(t) = \frac{\text{cov}\left(C(\lambda_1,t), C(\lambda_2,t)\right)}{\sigma(\lambda_1,t)\sigma(\lambda_2,t)},$$

where cov is the covariance, $\sigma$ the standard deviation and $C(\lambda,t)$ the meridian scan for a given wavelength, $\lambda$, and time, $t$. The value of the correlation coefficient is limited between $-1$ and 1 inclusive, where 1 indicates that the $C(\lambda_1,t)$ and $C(\lambda_2,t)$ are identically shaped and proportional to each other, and $-1$ indicates that they are $180°$ out of phase. A value of 0 indicates that their relationship is completely random or $90°$ out of phase. The resulting correlation is independent of amplitude (normalised), ensured by the standard deviation divisors in the equation. It is also symmetric, meaning that $\rho_{\lambda_1,\lambda_2} = \rho_{\lambda_2,\lambda_1}$.

Figure 7 shows the calculated correlation coefficients. In order to understand the coefficients, we first have to take into consideration the dynamics and geometry of the MSP measurements. The different measured auroral emissions have substantially different emission lifetimes, with the shortest being the prompt emission of the 427.8 nm, which has a lifetime of $5.38 \cdot 10^{-8}$ s (Itikawa et al., 1986). The other two are significantly slower at $\approx 0.7$ s for the 557.7 nm emission and 110 s for the 630.0 nm emission. It is worth noting that ionospheric density changes and quenching could shorten these lifetimes somewhat. Since the scan time for the MSP is 16 s, both the emissions at 427.8 nm and 557.7 nm will persist in only one scan. In the case of the 630.0 nm emission, it will be visible over several scans, blending together recent emissions with previous emission, as seen in figure 7. In relation to the correlation, this means that in most cases the correlation between the 630.0 nm emission and the other two will be largest at the start of the auroral activity.

With respect to the calculation, there are some geometrical considerations to take into account. As the MSP scans from magnetic north to magnetic south, a given angle off magnetic zenith will intersect different field lines depending on the altitude





**Figure 7.** The top figure shows the MSP data just before and after the 07:34 UT launch. The vertical blue stripe is due to clearing of snow from the slit. *L* and *KHO* mark the launch and passing east of KHO, respectively. Dashed lines represent the sunlit part of the atmosphere, with everything south of the line being sunlit at and above that altitude. Sunlit means that the light from the Sun has not passed below 150 km in order to get to the point of interaction. In the 630.0 nm figure the dashed and dotted gray line represents the rocket track mapped to the MSP field of view at a height of 240 km. The bottom figure shows the correlation coefficients for the time around the launch at 07:34 UT based on the data in the top figure.





one is looking at. This means that if an emission occurs on a given field line, the 630.0 nm emission from that field line will
end up in a different angular bin, compared to the 557.7 nm emission, due to the difference in emission altitude. This effect is
smallest close to zenith, and becomes larger as one moves away from zenith. Due to this change in angle, one would expect a
reduction in the correlation coefficient between these two emission, as the shape of the angular scan for the two emissions is
different.

Taking all of these considerations into account, it is now possible to start analysing the correlation coefficients in figure 7.
Firstly, it can be noted that the correlation between the red (630.0 nm) and the green (557.7 nm) emissions corresponds to
periods of strong auroral activity, showing correlation values larger than 0. At certain points in time the red-green correlation
is closely followed by the green-blue (427.8 nm), and the red-blue, for instance at 07:27 UT, 07:36 UT (120 s) and 07:44 UT
(600 s). One type of event that can generate this similarity is a period of high energy precipitation around magnetic zenith,
which causes emissions from the three lines within the same scan. These events are well known.

Looking at the correlation at 07:38 UT (240 s) and 07:50 UT (960 s) we have a situation where the red-blue correlation
is smaller than the green-blue. Taking into account the previously discussed mechanisms, the blue emissions occur at similar
altitudes to the green emissions. By following the correlation coefficients, it is possible to see that after two to three minutes
the difference in the correlation coefficient switches, with red-blue correlating more than green-blue. From the MSP data in
figure 7, we can see that if the emissions are from altitudes in the range 300–400 km, we can expect them to be sunlit. The
sunlit altitude is displayed in the figure, calculated according to the method described in subsection 2.2. This observation of the
time delay is consistent with the "cooking time" responsible for the observed ion upflow in the ionosphere (Lund et al., 2012).
The ion upflow drags with it neutral molecules from lower altitudes, and would subsequently increase both the availability of
$N_2$ and $N_2^+$ for the generation of sunlit aurora. An increase in available nitrogen was further confirmed by the observation of *in*
*situ* 427.8 nm emission onboard the RENU2 rocket (Godbole et al., 2019). Together with the rate calculations in the previous
section, we argue that a significant part of the observed 427.8 nm emissions can be attributed to sunlit aurora, which is further
backed up by the observations made from the photometer onboard RENU2.

## 5  Conclusions

We have presented an optical study of the sunlit aurora present in the time around the RENU2 launch, 13th of December 2015.
Calculations of the possible EUV height show that the upper parts of the ionosphere above the Kjell Henriksen Observatory
were sunlit at the time. An analysis using the correlation coefficient between three different wavelengths (427.8 nm, 557.7 nm
and 630.0 nm) measured along the magnetic meridian at a given point in time was used to demonstrate the detection of sunlit
aurora from ground-based instrumentation. By utilizing this correlation we show that the 427.8 nm emission occurs at the same
heights as the 630.0 nm emissions. Using data from the RENU2 rocket and the estimated solar photon flux, it was found that
sunlit aurora was a major part of the observed 391.4 nm and 427.8 nm emissions. The EUV photon flux and the electron flux
were estimated to be $j_\gamma \approx 8.3 \cdot 10^{10}$ cm$^{-2}$s$^{-1}$ and $j_e \approx 1.2 \cdot 10^{10}$ cm$^{-2}$s$^{-1}$ respectively. Using these fluxes the ionisation rate
of $N_2^+$ by photons was found to be $q_\gamma = j_\gamma \cdot \sigma_\gamma = 1.7 \cdot 10^{-6}$ s$^{-1}$ and by electrons $q_e = j_e \cdot \sigma_e = 2.4 \cdot 10^{-6}$ s$^{-1}$. We have also





shown that the correlation method presented here can be used to understand the altitude of the 427.8 nm emission as well as some of the processes occurring.

*Author contributions.* Article is primary written by Pål Gunnar Ellingsen with input from the other co-authors. Marc Lessard was additionally the RENU 2 PI.

*Competing interests.* The authors have no competing interests.

## 6 Acknowledgments

We acknowledge the Space Physics Data Facility(SPDF) and SPDF Contact: Natalia Papitashvili <Natalia.E.Papitashvili@nasa.gov>
for the use of OMNI data which was downloaded from http://lasp.colorado.edu/home/eve/data/data-access/. RENU 2 data are available at NASA's Space Physics Data Facility, https://cdaweb.sci.gsfc.nasa.gov/index.html/. Supported by the Research Council of Norway/CoE under contract 223252/F50. Support was provided at the University of New Hampshire by NASA ward NNX13AJ94G. JHH wishes to acknowledge support from NASA grant NNX13AJ93G. The data from the photometer plot can be found at http://mirl.sr.unh.edu/projects_renu2/FlightData/Hecht/





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
