# Peer review of "Observations of sunlit $N_2^+$ aurora at high altitudes during the RENU2 flight"

_Annales Geophysicae, 2020_

## Referee Comment (RC1) · Anonymous Referee #1 · 19 Aug 2020

Observations of sunlit N2+ aurora at high altitudes during the RENU2 flight
Pål Gunnar Ellingsen et al.
MS No.: angeo-2020-50

The paper is a study of near coincident data from meridian scanning photometers (MSP) and instruments on board the RENU2 rocket. The time of flight around 07:30 UT over the Svalbard region meant that the rocket passed through the daytime auroral region when the higher regions of the atmosphere were still sunlit. The role of the sunlit ionosphere under auroral conditions is indeed an important topic, and the effect of the resonance scattered emissions must be accounted for, especially when N2+ measurements are used for estimating fluxes and energies of the precipitation. However, the paper as presented does not add significant results to the subject, and some of the methods used and conclusions reached require more rigorous examination.

The MSP scans intercepted several poleward moving arcs during the time around and during the rocket flight. The instruments on RENU2 include a photometer measuring the N2+ emission at 391.4 nm and an electron plasma sensor (EPLAS). The paper uses results from the rocket data, which include modelling of the emissions, and also the ground measured emissions from three MSP channels. The latter include the N2+ emission at 427.8 nm.  Both 391.4 nm and 427.8 nm emissions are from the First Negative bands of N2+ and are the result of excitation which occurs simultaneously with ionization on particle impact.  However, the First Negative bands of N2+ are also excited by solar photons. This is a two-stage process in which the N2+ ions are produced by electron impact and by photoionization. Ground state nitrogen ions then act as very efficient resonant and fluorescent scatterers of sunlight, producing radiation at 391.4 nm and 427.8 nm. The paper is an exploration of this now established physics of the auroral sunlit atmosphere. There are several strands to the work, which do not add together very coherently as presently written.

It is stated (line 30) that one focus is to study the link between sunlit aurora and upwelling.  Apart from establishing that sunlit aurora was present and hence an extra source of N2 was needed at high altitudes, the assumptions made in the correlation analysis of MSP data are dependent on many uncertainties, which are not quantified. Another stated focus is to study how different processes contribute to N2+ emissions. In this regard, the final sentence of the abstract is misleading unless I have misunderstood. Does it refer to the figure of 40% derived in Section 4.1.3? This rough estimate of the ionisation rates from both photoionization and electron impact confirms the known fact that their ionization efficiencies are very similar and therefore very hard to distinguish. It is not clear what this calculation adds to the study. However, the statement in the abstract claims that sunlit aurora was a major part (40%) of the observed 427.8 nm emission. Indeed such estimates have been made before by several researchers from Svalbard observations. But the separation of the ionization sources is not the same thing.

Figure 4 shows the data from the EPLAS; these raise some questions when comparing them with the ground photometers and the all-sky camera images that are included as supplementary data. The characteristic energies are very small (~100 eV), but with quite large fluxes of electrons during short bursts. Such electron data are not directly comparable with what is observed from the ground along the magnetic meridian. So how does the rocket trajectory coincide with the all-sky images? Although the all-sky camera data are very integrated in time, could an image or two be used to clarify where the rocket's footprint is, and what it may be measuring in relation to the optical aurora during the times of interest?

The aurora seen at 07:40 is bright and structured, to the south of KHO, and moves rapidly northward, seeming to be overhead at KHO at 07:42:22. Cloud is then an issue but it looks like a very narrow arc is overhead at KHO at 07:44:50. Then another bright arc forms south of KHO. The rest of the interval seems cloudy and with much less structure in the aurora, although it may be obscured. The MSP 5577 has 4 kR brightness in the arc to the south of zenith and from the auroral images of this active rayed arc it is clear that the energies must be far greater than the peaks of 200 eV measured on the rocket. No mention of this discrepancy is made. An obvious question that arises is how closely does the aurora seen from KHO relate to the EPLAS measurements?

The next part of the study (4.1.5) is a comparison between the rocket measured 391.4 nm emission, and a modelled estimate, in order to demonstrate that the emission measured by the rocket is from the resonance scattered component of the emission.  It is essential to describe in detail the procedure and assumptions required to produce figure 6, rather than rely on the Hecht et al 2019 paper. Many questions arise about the photometer field of view, the model spectra used and what exactly is included in the output model curve.  How are photoelectrons included as an input source in the model? Is the model result shown purely 391.4 nm emission? The caption states it is dayglow plus emission due to auroral precipitation, but the axis label is 391.4 nm brightness. At these heights and at such low energies there will be minimal contribution from other emissions, but some discussion of these important factors is needed.

It is stated (line 194) that the resonance scattered contribution to the emission vastly exceeds that from direct excitation. Although this statement is not surprising, where is the evidence? Is it the result of modelling? The scattered contribution is dependent on shadow height, height of observation and energy of precipitation. How all of these factors affect the statement is important to know. At line 198 is it stated 'there are clear increases that correspond to model predictions ...' ; however, these correspondences are not clear, so more direct evidence is needed of these instances.

It is claimed (line 200) that the most noticeable aspect of the data is the increase in emissions after 500 seconds, which is **not** reflected in the model. If there is indeed a clear difference between the data and model during this time it needs to be presented more clearly so that the obvious nature of the feature is evident. At this time there is a large increase in the flux of low energy electrons, so an increase in brightness of emissions which result from low energies might be expected at these heights. If there is indeed no increase in 630.0 nm emission at this time, how is this explained? A breakdown of the model results is of importance for any conclusions that are drawn. On reading Hecht et al 2019, there appears to be some increase in both 630.0 nm and 557.7 nm emission from the model at this time, so the authors need to provide more details in the present paper. However, given the height and position of the rocket, the most likely emission would indeed be the resonant scattered 391.4 nm.

The correlation analysis of MSP data is discussed in terms of issues with geometry, which are indeed significant. But it is not obvious exactly **how** all these very serious matters have been taken into account. Where is the crucial information about the analysis of the raw photometer data? How have the profiles of the scans been integrated? How has a background been subtracted? Looking at individual scans, is the scattered contribution visible as an increase in brightness above the shadow line at any time, especially when scanning through an arc? Interpreting such data, however, would be complicated by the occurrence of more than one arc in each scan. And if there are several auroral features within each scan how is the brightness measurement interpreted?  The only angle where the true comparison can be made about the relative brightnesses at any time is in the zenith. Since

this is not the case here, more justification for the method in the light of the inherent uncertainties is needed.

The main conclusion drawn from this analysis appears to be the seeming delay between different correlations, and the interpretation of this delay as an upflow of ions. The authors also claim that the correlation analysis shows that the height of the 427.8 nm emission is the same as the 630.0 nm emission. The data do not justify such conclusions without much more rigorous analysis and interpretation of the events at the times when the correlation 'switch' is observed. The uncertainties in the correlations need to be assessed quantitatively.

A tantalising reference is made in the final section to RENU2 measurements of 427.8 nm emission and a reference to a paper unavailable (Godbole et al, under review).

Minor points

64 Poleward

71 this emission isn't relevant

103 reference to the figure isn't needed here

104 are available

125 missing stop

147 clarify, what ionisation rates? Confusing since the aim is to find the ionisation rate

166 sentence not clear. Removing the first 'as' would help

Same paragraph - it is not clear what the point of the last two sentences is here.

172 Not included in what?

187 'somewhat' needs enlarging

191 - and produced by solar photons?

198 'could affect' - explain more

---

## Author Comment (AC1) · 8 Sep 2020

UiT The Arctic University of Norway
PO Box 385
8505 Narvik

September 8, 2020

Referee #1
Annales Geophysicae

Dear Sir or Madam

Thank you for taking the time to review our submission *Observations of sunlit $N_2^+$ aurora at high altitudes during the RENU2 flight* . This response will reference each paragraph individually.

**Paragraph 3** The referee is correct in assessing that the stated 40% is from the calculations in 4.1.3, though that calculation uses electron precipitation measurements taken by the rocket. Doing such a calculation with only ground data would be more speculative, as the input electron precipitation would not be known.

**Paragraph 4** The track of the rocket is projected onto the MSP data in figure 7 (we acknowledge that figure 7 contains a lot of data). As the rocket flew east of the observatory as shown in figure 1, projecting that path onto all-sky images would introduce some geometric affects. A rough position in the images could be shown in the video, though as the camera is not geometrically corrected, the position would be uncertain within some degrees. Additionally as the referee correctly states, the rocket (due to its uncertainties during launch), did not fly close to magnetic zenith for the observatory, making it harder to determine which field lines in the All-sky image it crossed. I would be hesitant to project the whole path on one or two images, as it would give an incorrect view of the aurora at any given point of the rocket pass.

**Paragraph 5**

We agree that there is clearly some discrepancy between what is seen in the MSP and what is measured by the rocket and this is expected, as the rocket flew east of the observatory. Therefore this paper does not try to directly relate MSP intensities to the measured electron energies, but rather looks into what the solar EUV contribution is to the nitrogen emissions (using space and rocket measurements). The ground measurements are used to look into the preceding effects and the hight distribution of the nitrogen emissions. These are then qualitatively compared to the measurements taken by the RENU2 rocket.

The ideal case would of course be that the rocket flew very close to magnetic zenith above the observatory, though this is hard to achieve with the current precision of the rockets in use. If a rocket (or maybe a cubesat) did fly there, it would allow for direct comparisons of emissions on the ground and electrons detected.

**Paragraph 6 - 8**

We will be adding some new text from line 183:

*The field of view of the photometers was 3.2 degree and the instrument was pointed in the zenith although the rocket spun and some glint was observed as described in Hecht et al. (2019). The photometer bandpass was narrow enough, at about 3 nm full-width at half maximum, to isolate the (0,0) band.*

*To model the data two long-established models were used, AURIC (Strickland et al., 1999) which completely modelled the dayglow and B3C (Strickland et al., 1993) to model the auroral output. For the $N_2^+$ emission resonant scattering of sunlight off $N_2^+$ ions dominated the emission for most of the flight.*

We still feel that the reader will need to read Hecht et al. 2019 for the details. The added text includes some details about the photometer including field of view. The two models are long-established and we have added references to their descriptions.

1. "The caption states it is dayglow plus emission due to auroral precipitation, but the axis label is 391.4 nm brightness."

   - Our answer is that these come from the outputs of the models

2. "At these heights and at such low energies there will be minimal contribution from other emissions, but some discussion of these important factors is needed."

   - We are a bit unclear on what is asked here (these plots are from model outputs):

3. Statement on line 194

   - The statement on line 194 is a result for modelling.

4. "The scattered contribution is dependent on shadow height, height of observation and energy of precipitation. How all of these factors affect the statement is important to know."

   - Again this is calculated by AURIC.

5. "At line 198 is it stated 'there are clear increases that correspond to model predictions ...' ; however, these correspondences are not clear, so more direct evidence is needed of these instances."

   - Between 550 and 600 s the model predicts increases in the green and blue signals that are due to strong auroral precipitation and are clearly seen in the figure.

6. "It is claimed (line 200) that the most noticeable aspect of the data is the increase in emissions after 500 seconds, which is not reflected in the model. If there is indeed a clear difference between the data and model during this time it needs to be presented more clearly so that the obvious nature of the feature is evident."

   - It appears very clear to us that there is a sharp increase in the 391.4 emission at 500 s that is not seen (say in the green emission) and not reflected in the model.

7. " At this time there is a large increase in the flux of low energy electrons, so an increase in brightness of emissions which result from low energies might be expected at these heights."

   - The model does not predict a measurable increase in either the green or blue emissions at 500 s.

8. "If there is indeed no increase in 630.0 nm emission at this time, how is this explained?"

   - The Hecht et al 2019 paper suggests that these are very short-lived bursts that are not seen in 6300 because of the long-lifetime of that emission.

9. "A breakdown of the model results is of importance for any conclusions that are drawn. On reading Hecht et al 2019, there appears to be some increase in both 630.0 nm and 557.7 nm emission from the model at this time, so the authors need to provide more details in the present paper."

- There is no increase in either the blue or green emissions at 500 to 510 s.

10. However, given the height and position of the rocket, the most likely emission would indeed be the resonant scattered 391.4 nm.

   - Yes we agree here.

**Paragraph 9**

1. Where is the crucial information about the analysis of the raw photometer data? How have the profiles of the scans been integrated? How has a background been subtracted?

   - 3.1 states that the measured valued background subtracted and averaged over two scans, and converted from counts to Rayleigh via calibration. In the analysis there is no integration. It is unclear what is meant with *raw photometer data*, the photometer gives out a voltage pulse which is collected by a counter, though this does not belong in the paper (as the function of an A/D converter in a CMOS camera chips doesn't belong).

2. Looking at individual scans, is the scattered contribution visible as an increase in brightness above the shadow line at any time, especially when scanning through an arc? Interpreting such data, however, would be complicated by the occurrence of more than one arc in each scan. And if there are several auroral features within each scan how is the brightness measurement interpreted? The only angle where the true comparison can be made about the relative brightnesses at any time is in the zenith. Since this is not the case here, more justification for the method in the light of the inherent uncertainties is needed.

   - If one was able to measure under each field line (i.e. having a chain of observatories) this could be determined. In the MSP data from one observatory, any attempt at doing this for individual scans would be plagued by overlap from different heights. One might expect to see something like this is a large statistical study with several years worth of data, which is outside of scope of this study. In this study what is looked at is the correlation between emissions for different wavelengths over a given scan. If the emission scans correlate, their "shape" is the "same" allowing for the conclusion that they should have been emitted within the same volume. When they don not correlate one can draw the conclusions that they come from different volumes.

**Paragraph 10**

It is unclear which rigorous analysis is missing. The analysis here is base on a statistical analysis which asses the uncertainties through the correlation coefficient. We do not understand what is meant with "The uncertainties in the correlations need to be assessed quantitatively" as the correlation coefficient is in itself a quantitative measurement of the uncertainties. A coefficient of 1 would mean a perfect correlation, reducing it towards 0 makes it less and less correlated (likely that they are the same).

**Paragraph 11**

We are hoping the Godbole paper is published soon. We can supply it to the referee if desired.

**Minor points**

**64** Can't see any difference

**71** Sentence removed

**103** Reference removed.

**104** Thanks

**125** Thanks

**147** Agreed, clarified to: *...λ = 50 nm based on the invariability of the photoionisation cross section from approximately 30 nm to 70 nm ...*

**166** Agreed. Changed to: *..., means that in solar EUV conditions one should be careful in classifying EUV as the only production mechanism for $N_2^+$, as an increase in the precipitating electrons can drive the production of $N_2^+$ as well.*

**172** Changed to: *A process not included in the estimation here,...*

**187** Removed somewhat as it was a poor choice of words.

**191** Added: *In the absence of precipitation dayglow emission is observed from resonance scattering from ambient N2+ ions produced by solar photons or photoelectrons but this is a minor portion of the observed emission.*

**198** Added: *As noted in Hecht et al. (2019) the short-lived nature of some of the intense precipitation could affect some of the observations of the auroral emissions. In particular the OI(630 nm) emission intensity was well below the model predictions. But emissions with short lifetimes (well below a second) are not affected.*

Kind regards on behalf of the authors

Pål Gunnar Ellingsen

---

## Referee Comment (RC2) · Anonymous Referee #2 · 13 Jan 2021

This is a well written and clear manuscript describing some aspects of the RENU rocket flight, in particular ground-based optical data and rocket-based particle data, to do with the sunlit aurora over the polar cusp. Evidence of neutral upwelling is found. Whilst no new major discovery is made, or problem solved, rocket in-situ data are rare and valuable and therefore should be published. On this basis, I recommend publication.

---

## Author Comment (AC2) · 13 Jan 2021

We want to thank the referee #2 for taking the time to review the paper and recommending it for publication.

---

## Author Response (AR2)

UiT The Arctic University of Norway
PO Box 385
8505 Narvik

June 25, 2021

Referee #1
Annales Geophysicae

Dear Sir or Madam

Thank you for taking the time to review our submission *Observations of sunlit $N_2^+$ aurora at high altitudes during the RENU2 flight* . This response will reference each paragraph individually.

- *From line 170: It is still not clear what 'not included' implies and it would be helpful if it was explained more clearly how this 'omission' is related to the ionisation rates estimated in this section, and the implications.*

    – We agree it was not completely clear. The paragraph has been rewritten.

- *Regarding the statement "The exact amount (of resonant scattering of sunlight) is hard to determine...", a quick search suggests that more recent work than Romick (2001) on this topic is Jokiaho et al. (Ann. Geophys., 27, 3465–3478, 2009), which is a statistical study from Svalbard. It shows how the proportion of scattered N2+ depends very strongly on both the energy of precipitation and shadow height. They state that this energy dependence means that case studies are more relevant than a statistical approach, given the many possible variations in auroral conditions. In the present work, the conditions over Svalbard during the rocket flight were quite variable. There is a discrepancy likely between the measurements (and model) from the rocket, and the brighter and more active auroral conditions within the MSP scans. Why not put the rocket track on the 427.8 nm MSP scans, and use this as a discussion point for the above comparison?*

    – The referenced paper is a study of remote data with a steady state model. As the paper and the reviewer also states, the conditions over Svalbard at the time of the rocket were anything but steady. Furthermore the referenced paper in its conclusion states its limitations and explains that the model is limited, which does not contradict the statement that "the exact amount is hard to determine".

    As for putting the rocket track on the 427.8 nm MSP scans (similarly discussed in "paragraph 4" of the previous response), it is put on the 630.0 nm track in figure 7, and could be copied to the 427.8 nm track, though we don't fell that it would add any value, as figure 1 shows that the rocket only crossed the track at one point. As these rockets are not steerable, the accuracy is not sufficient to guarantee a flight along the MSP line (which would not be possible anyway due to the location of the launch pad). The error, if the MSP data was directly compared with the rocket, would be large given the discrepancy in position. This was also discussed in the paragraph 4 in the previous response.

    We have rewritten the paragraph and included an explanation of why the comparison is not done.

- *There is still a question (see previous paragraph 3) about the statement: "..it was found that sunlit aurora was a major part ( 40%) of the observed 427.8 nm emission." Section 4.1.3 provides an estimate of the ionisation rates from both photoionization and electron impact, and so it confirms the known fact that their ionization efficiencies are very similar and very hard to distinguish (also stated by Jokiaho, 2009). The real significance of the 40% figure for 427.8 nm emission that is novel and of importance, needs to be made clearer.*
    - The significance of the estimation comes from the fact that it uses *in situ* data, compared to the indirect remote sensing data as input to a static model in Jokaiho et al. 2009. To make it clearer, *in situ* has been added to the abstract and *found* changed to *estimated*.
- *Discussion of Fig.6 (wrongly named 4.1.5 in text around line 200) is still hard to follow, not helped by some unintended repetition of a sentence within this paragraph (lines 201 and 204). The statement "The resonance scattering emission vastly exceeds the direct auroral emission" is followed by a sentence with "therefore..." but the logic of this sequence is unclear. The sentence that is repeated is vague - "could affect some of the observations." I presume the sentence should be in its second position, leading on to the discussion of 630.0 nm. The paragraph needs to be rewritten to be a logical entity.*
    - Sorry the wrongly labeled reference, it was due to an obscure LaTeXerror. We are also sorry for the confusion by the duplicate sentences. We have rewritten this paragraph, see the difference document.
- *The comparison of the data and the model is still not easy using this figure. The answers to the previous review do not attempt to help this situation. Having to compare the presented results with those from the published Hecht paper is not ideal. I wonder if some helpful labels or shaded regions could be included to aid the eye. If I have followed correctly, it is claimed that there is a good comparison with the model between 550-600 from precipitation induced ions which have scattered emission, but between 500-640 the 'plateau' is not seen in the model, hence it must be from extra N2+ ions from upflows in the precipitation region. If I am correct in my interpretation, then these features would be easier to see if they were shaded or marked in some way, and then the text made more clear. I have had to read the two paragraphs many times while examining the figure to check I was seeing the features as described.*
    - The paragraph has been rewritten in an attempt to clarify this better. See the difference document for the changes.

Kind regards on behalf of the authors

Pål Gunnar Ellingsen

---

## Author Response (AR3)

UiT The Arctic University of Norway
PO Box 385
8505 Narvik

August 9, 2021

Topical editor
Annales Geophysicae

Dear Daniel Whiter

Thank you for the comments on our submission *Observations of sunlit $N_2^+$ aurora at high altitudes during the RENU2 flight.* I am confident that the replies to the comments listed below are sufficient to get the paper accepted.

- *Line 120: 428.7 should be 427.8.*
    - Thank you, is has been corrected
- *There is still a date with an ordinal suffix, on line 139 ("13th"). Please convert this to a date (with month and year) both to avoid ambiguity and to follow the journal guidelines.*
    - It has been changed
- *Line 176: I assume "comprehensible" should be "comprehensive".*
    - Changed
- *Figure 6 caption: Please add "(dashed line)" and "(blue line)" into the caption somewhere to make things easier for the reader and to avoid ambiguity.*
    - It has been added to the text even though the figure legend contained the information.
- *Figure 6: Please mark on the start and end times of the "plateau", e.g. with shading, vertical lines, or arrows, as requested by the reviewer. This plateau is unclear to the reviewer, so is likely to also be unclear to other readers.*
    - I am sorry to say I can not see any good reason for cluttering the figure with annotations. The text clearly references the data and time points (500 s. and 630 s), which can easily be found on the $x$-axis. Throughout the article there are numerous references to axis coordinates without any annotations. This is common practise in scientific papers. Might I speculate that the reviewer forgot that the $y$-axis is logarithmic, and thereby missing the significant step up/down at the referenced time points?
- *Line 206, sentence starting "However": I suggest rewriting this sentence to make it easier to read. Perhaps this is an accurate rewrite: "However, during this period the AURIC/B3C modelling shows that the resonance scattering component of the 391.4 nm emission vastly exceeds the direct auroral emission component, and is enhanced due to the increased production of N2+ ions by the auroral precipitation."*
    - The sentence has been rewritten and simplified.
- *Line 216, sentence starting "The data": I'm sorry but I don't understand what point you are trying to make with this sentence, and I don't think the reviewer did either. Are you saying the model predicts that emission from auroral precipitation should be observed from 450s onwards? If so, what point does the "therefore" follow from?*

– The sentence has been rewritten and simplified.

- *Line 269: "is closely followed by" - to me this reads as if there is a lag between the correlation coefficients, but I don't think that's what you mean. Probably "closely follows" is better.*

  – You are correct. Thank you.

Kind regards on behalf of the authors

Pål Gunnar Ellingsen